# Clinical Evaluation of Pathognomonic Salivary Protease Fingerprinting for Oral Disease Diagnosis

**DOI:** 10.3390/jpm11090866

**Published:** 2021-08-30

**Authors:** Garrit Koller, Eva Schürholz, Thomas Ziebart, Andreas Neff, Roland Frankenberger, Jörg W. Bartsch

**Affiliations:** 1Department of Neurosurgery, Philipps University Marburg, Baldingerstr, 35033 Marburg, Germany; garrit.koller@kcl.ac.uk (G.K.); evaschuerholz@aol.com (E.S.); 2Centre for Host-Microbiome Interactions, Faculty of Dental, Oral and Craniofacial Sciences, King’s College London, Guy’s Tower, Floor 17, London SE1 9RT, UK; 3Department of Operative Dentistry and Endodontics, School of Dentistry, Campus Marburg, Philipps University Marburg and University Medical Center Giessen and Marburg, Georg-Voigt-Str. 3, 35039 Marburg, Germany; frankbg@med.uni-marburg.de; 4Department of Oral and Maxillofacial Surgery, Campus Marburg, Philipps University Marburg and University Medical Center Giessen and Marburg, Baldingerstr, 35033 Marburg, Germany; Thomas.Ziebart@uk-gm.de (T.Z.); mkg@med.uni-marburg.de (A.N.)

**Keywords:** Oral Biomarkers, caries detection, real-time protease activities, metalloproteases, FRET-peptides, proteolysis, extracellular matrix degradation, protease inhibition

## Abstract

Dental decay (Caries) and periodontal disease are globally prevalent diseases with significant clinical need for improved diagnosis. As mediators of dental disease-specific extracellular matrix degradation, proteases are promising analytes. We hypothesized that dysregulation of active proteases can be functionally linked to oral disease status and may be used for diagnosis. To address this, we examined a total of 52 patients with varying oral disease states, including healthy controls. Whole mouth saliva samples and caries biopsies were collected and subjected to analysis. Overall proteolytic and substrate specific activities were assessed using five multiplexed, fluorogenic peptides. Peptide cleavage was further described by inhibitors targeting matrix metalloproteases (MMPs) and cysteine, serine, calpain proteases (CSC). Proteolytic fingerprints, supported by supervised machine-learning analysis, were delineated by total proteolytic activity (PepE) and substrate preference combined with inhibition profiles. Caries and peridontitis showed increased enzymatic activities of MMPs with common (PepA) and divergent substrate cleavage patterns (PepE), suggesting different MMP contribution in particular disease states. Overall, sensitivity and specificity values of 84.6% and 90.0%, respectively, were attained. Thus, a combined analysis of protease derived individual and arrayed substrate cleavage rates in conjunction with inhibitor profiles may represent a sensitive and specific tool for oral disease detection.

## 1. Introduction

Together, periodontal disease and caries comprise the most prevalent chronic conditions of infectious aetiology, with 3.6 billion people worldwide affected by caries alone in the permanent dentition [1,2]. In the United States, the population prevalence of untreated dental decay is estimated at 27% within the dentate adult population, with more active and severe disease in some population groups [3], as well as being associated with overall health inequality [4]. Active caries is a progressive pathological process of combined, acid-mediated dissolution of the calcified tooth structure and degradation of the proteinaceous matrix of dentine. However, carious lesions may not be actively progressing due to shifts in the host or microbial composition. Caries conceptually shares features with periodontal disease with both, the respective ECM and hard tissue degradation, and is attributed to the combined effect of host and microbial contributors. For dental decay, this has led to a tissue-specific hypothesis [5]. Previous studies examined the presence of individual members or classes of proteases in oral disease and its progression, predominantly focusing on host proteases. These studies have focused on metalloproteases and cathepsins, with significant contributions ascribed to gelatinolytic or collagenolytic MMPs. Bound within dentine, the matrix metalloproteases enamelysin (MMP-20), collagenase (MMP-8), the gelatinases A (MMP-2) and B (MMP-9), and cysteine proteases cathepsins (B and K) have been functionally shown to participate in dentine matrix destruction [6,7,8,9,10] and linked to the adhesive failure of composite restorations [11]. In addition to host-derived enzymes, microbial proteases have been implicated in the process, such as the recently identified U32 protease family in caries [12] and gingipains, comprising papain-like serine proteases in periodontal disease. Many of the participating proteases in the caries process have also been described in inflammatory bone destruction in periodontitis [13]. To date, a few studies have examined the aggregate, host, and microbially-derived network of proteolytic activity, as well the potential use of such networks as disease and activity markers. Such findings may address the significant unmet clinical need for easily accessible and readable disease markers for screening, diagnosis, or disease monitoring for periodontal disease and caries.

Despite many efforts to identify proteomic or genomic candidate biomarkers, these have not yielded clinically accessible disease markers. For enzymes, this is aggravated by the inability to causally link protein quantity to pathophysiology, as biological, enzymatic activity is not ascertained.

This multi-level regulation is highlighted by MMP proteases, for which mRNA expression levels do not directly relate to protein synthesis or subsequent extracellular localization. If MMPs are secreted, their activity is tightly regulated by sequential activation, co-factor dependency and active proteases are readily inactivated by physiological inhibitors, such as TIMPs and α-2-macroglobulin. Within saliva, the predominance of pro-forms or inactivated MMPs has been reported [14], thus mandating readouts of activity rather than concentration if specific enzymes are to be functionally linked to tissue destruction. Saliva contains diverse, multi-class host proteases, including cysteine cathepsins CTSB, CTSC, CTSV, CTSX/Z/P, the serine cathepsin CTSA and the aspartic cathepsin CTSD, kallikreins KRK5, KRK6, KRK7, KRK10, KRK11 and metalloproteases MMP-1, MMP-3, MMP-7, MMP-8, and MMP-9, as well as the serine proteases dipeptidyl peptidase (DPPIV/CD26) and chymotrypsin-like proteinase 3 (PRTN3). Whilst most major host proteases described are shared between serum and saliva, the differences in health are delineated in saliva by the absence of ADAMTS-13, MMP-3, proprotein convertase 9 (PCSK9) and uPa/urokinase (PLAU), and presence of CTSV, KRK5, KRK10, KRK11, MMP-7 and MMP-8.

Saliva has attracted interest in diagnosing and screening a wide range of applications and, as a readily accessible biofluid, may enable the analysis of specific patterns of protease activity derived or altered by oral diseases, including caries. However, studies have shown that diverse and poorly-inhibitable proteases activities present in saliva lead to the rapid degradation of many candidate protein biomarkers, including phosphoproteins involved in the maintenance and repair of mineralized tissues, histatins, proline-rich proteins and statherin [15]. However, proteolytic enzymes involved in the disturbance of tissue integrity or repair might serve as functional and diagnostic markers for oral diseases. Measuring the combined (host and microbial) activities against arrayed collagenase/gelatinase substrates may overcome issues arising from the diverse, changing oral environment and provide an overall phenotypic fingerprint of activity capable of tissue-destructive processes, rather than relying on the quantitation of multiple potential effector proteins.

The present proof-of-concept, cross-sectional clinical study utilised a sensitive, multiplexed approach for the simultaneous detection of protease activities, including matrix metalloprotease and ADAM (A Disintegrin And Metalloproteinase), many of which are known to effect ECM destruction [16,17]. This comprised a combined experimental and mathematical method, based on time-lapse fluorescence measurements of a panel of moderately specific FRET-based peptides. Whole mouth saliva (WMS) samples of patients with caries periodontal disease cases, as well as healthy controls, were evaluated. Where available, matched caries tissue biopsies were included to correlate the potential analyte matrix (WMS) activity to the disease process (biopsy material). Multi-class endopeptidase activities specific to dental decay were examined in small volumes of whole mouth saliva samples to overcome current diagnostic constraints. Using the sensitivity of fluorescent enzyme substrates (with amplification arising from many individual substrate molecules cleaved by one single, active protease) and the combined specificity of multiple protease substrates within the array, delineated unaffected, steady-state endopeptidase activities and those affected by disease state.

## 2. Materials and Methods

### 2.1. Recruitment of Patient Cohort and Collection of Saliva Samples

Patients attending the Department of Oral and Maxillofacial Surgery of the University of Marburg and undergoing elective extraction of teeth were invited to participate in the study. Informed, written consent was obtained from all patients, in accordance to the Helsinki Declaration. Following intraoral and radiographic examination, patients were dichotomized based on the presence or absence of cavitating carious lesions affecting dentine, periodontal disease or healthy, control patients. Other dental or medical conditions were recorded. Details regarding the patient cohort are provided in Table 1. This pilot study obtained ethical approval from the local Ethics Committee (Marburg University, Marburg, Germany; Registration Number No. 29/17). Informed consent was obtained from all patients to use their biological specimens and clinicopathological data for research purposes. Details of the general medical history of the patients were recorded.

Fifty-two consented patients provided unstimulated WMS by passive drooling. Where extractions were non-surgical, and the crown remained intact, extracted teeth were included in the study. Immediately upon sampling, specimens were snap-frozen in liquid nitrogen until further analysis. Seven WMS samples were used for method validation, and the remaining 45 saliva samples (with active caries *n* = 32, caries-free *n* = 13) were used for downstream analysis. For 22 cases, caries biopsies were obtained from cavitated, extracted teeth. Here, four samples were used for method validation and excluded from downstream analyses.

### 2.2. Determination of Proteolytic Activities in WMS and Caries Samples

Protease activities were examined by a modified Proteolytic Activity Matrix Analysis (PrAMA) technique (Miller et al., 2011). Multiplexed and arrayed FRET-polypeptides, structurally based on physiological ECM-protease targets, were used. Upon successful cleavage, a fluorescent probe (5-Carboxyfluoresceine) is de-quenched and can be monitored by fluorometry. These comprised: substrates PepA (PEPDab005, all BioZyme Inc, Apex, NC, USA), PepB (PEPDab008), PepC (PEPDab010), PepD (PEPDab013), and PepE (PEPDab014) were used, based on physiological targets, including gelatinase/collagenase substrates. The provision of dually quenched peptides confer resistance to ubiquitous carboxy- or amino-peptidases [18], particularly evident in salivary samples. PepA-PepE varied in their specificities and activities towards different metalloproteases (MP) belonging to MMP or ADAM families. In addition to the uninhibited multiplexed assays to assess activity against substrates used, parallel inhibitor groups were included to identify the absolute and relative contribution of protease family or classes relative to the overall levels of proteolysis observed. Inhibitor groups comprised the high affinity, irreversible inhibitor of MMP, Batimastat (BB-94; MMPi group), the complete™ inhibitor cocktail (Roche, Penzberg, Germany) targeting calpains, serine and cysteine (CSC) proteases (COMPi) group, or both inhibitors (BOTHi group). Positive and negative controls comprised wells with 0.01% (*w*/*v*) trypsin and FRET-substrate in activity buffer, respectively. Substrate identities, sequences, known and predicted target enzymes are provided in Table 2. The modified PrAMA-based analysis was performed as described earlier [16,17] with modifications. Briefly, for time-lapse fluorimetry, 10 pmol substrate/50 μL of activity buffer (1 μM ZnCl_2_, 20 mM Tris-HCl pH 8.0, 10 mM CaCl_2_, 150 mM NaCl, 6 × 10^−4^% Brij-35) was used.

Immediately after thawing, WMS and ultrasonically homogenized caries tissue were spun at 13,000 g for five minutes. The supernatants were used for WMS at 1:20 (*v*/*v*) and for biopsy tissue at 1:750 (*v*/*v*) dilutions in activity buffer. Plates were read every 5 min for 1 h using a fluorescence plate reader at 37 °C (BMG Fluostar Optima, Offenburg, Germany), using excitation and emission wavelengths of 485 and 530 nm, respectively. From the data obtained, the signal of negative control was subtracted. A linear, four-point curve fitting model was used to determine the maximum cleavage rate (Vmax) and, accounting for substrate depletion and photobleaching decay, to determine turnover rates. Cleavage rates of peptides in the UMWS and caries biopsies were determined and analyzed as absolute (a.u./min) and relative activities. Absolute activities were determined from maximum kinetic activities (expressed as Vmax, a.u./min), and relative cleavage rates between substrates or inhibitor groups in each sample (percentage relative to other substrates or the uninhibited control). were obtained. Experiments were conducted at the same time for all sample conditions.

### 2.3. Cleavage Rate Analysis and Statistics

For the cleavage rate analysis, statistical significance was determined using a *t*-test with a threshold for significance determined at *p* = 0.05. For cases with unequal variance, the Welch test was used for pairs. Values are denoted as not significant (ns, *p* ≥ 0.05), significant * (*p* ≤ 0.05), highly significant ** (*p* ≤ 0.01), or very highly significant *** (*p* ≤ 0.001). Effect sizes were estimated by Cohen’s D (Threshold of 0.2, 0.4 and 0.6 denoting small, medium, and large effect sizes). To determine the effect of multiplexed substrate cleavage profiles, overall and relative activities, degrees of protease inhibition, and patients’ disease status, after initial visual analysis of descriptors, a multivariable analysis was conducted using a supervised machine learning approach with best-fit approaches (Leclerq et al., 2018). To reduce dimensionality from the multiparametric cleavage rates and inhibition profiles obtained from patients before bootstrapped data sampling and cross-validation using kNN, Logistic regression, Neural Network, a forward-pruning tree algorithm and a stacked combination of all. Classifications were carried out using BiodiscML [19] and Orange3 [20]. Statistical analyses were conducted using JASP (JASP Team (2020), Version 0.13, JASP Team, Amsterdam, The Netherlands).

## 3. Results

### 3.1. Biochemical and Protease Profiles in Saliva

Before dilution, the salivary protein concentration mean was 10.35 mg mL^−1^ (C.I. = 7.47–13.94). The mean caries biopsy mass obtained was 3.9 mg (C.I. = 1.7–7.9), with variable total protein concentrations derived from these and average protein contents of 370.6 µg (C.I. = 42.5–825), before final dilution. The total protein of 12.5 µg of caries and 500 µg saliva were present per well (1:40 ratio). All five peptides were processed by salivary peptidases to different degrees and demonstrating consistent and differential protease activities between subjects and sample types (Figure 1).

In terms of overall kinetic activity, PepC and PepE were the most effectively cleaved substrates consistently across all saliva samples (Figure 1 Columns 3,5), representing 46.5% (CI 29.8–61.8; V_max_ = 2140.7, SD 1487) and 22.7% (C.I. 11.4–39.4; V_max_ = 1246.3, SD 1275) of the combined arrayed activity, respectively. PepA, PepB and PepD accounted for 13.5% (C.I. 5.1–22.2; V_max_ = 552.7, SD = 459), 9.8% (C.I. 5.8–17.7; V_max_ = 465.1, SD 390) and 7.4% (C.I. 2.3–14.9; V_max_ = 335.5, SD 352), respectively. Detailed cleavage data is provided in Appendix A
Table A1. A consistent, trypsin-like CSC activity in saliva was demonstrated by PepC COMi (Figure 1 Column C and 2), suggesting substantial cysteine cathepsin and kallikrein activity. However, the MMP activity and inhibition levels were variable between substrates and subjects. This finding suggested differential contributions of activities from MMPs such as MMP-1, MMP-3, MMP-7, MMP-8, MMP-9 and potentially MMP-20, and their contribution relative to other CSC or other (non-MP and non-CSC) proteolytic signatures. These residual activities that could not be inhibited by either MMPi, COMi or BOTHi, represented the most significant contribution to the total cleavage rates observed overall. From these data we postulate the presence and activity of different protease families and groups, with large proportions of activity observed not attributable to MMPs, serine or cysteine endopeptidases. As high-turnover and pan-MMP substrates, PepB and PepA displayed significant MMPi inhibition across all saliva samples, with residual activity of PepB at 74.0% (CI 54.3–87.7) and 78.9% for PepA (CI 49.2–97.3) of the respective uninhibited kinetic activity. PepD had an average residual activity at 91.4% (CI 70.6–126.9), PepC 87.4 (CI 73.4–104.6). Despite effective cleavage of PepE, the substrate demonstrated poor overall inhibition within the MMPi group, with slightly increased average activity of 102.5% (CI 90.0–129.2), relative to uninhibited samples. Detailed cleavage data is provided in Appendix A
Table A1.

The average CSC (COMi) and combined CSC/MP (BOTHi) contribution to kinetic activity was approximately 25% against the arrayed peptides. The CSC contribution was, on average, 33.7% for PepA (SD 18.6; C.I. 6.0–61.9); 18.9% for PepC (SD 14.9; C.I. 2.2–48.5) and 17.4% for PepB (SD 10.1; C.I. 4.8–35.1), and 14.0% for PepD (SD 8.2; C.I. 3.1–26.7). PepE was not consistently processed by CSC proteases present, with inhibition levels of 7.7% (C.I. 1–17.7) observed.

Overall, the substrates arrayed demonstrated differential, but also demonstrated some consistency of utilisation between subjects. PepA was effectively cleaved with a high relative turnover by both CSC and MP proteases. PepB had a higher propensity to MMPs s. pepC showed a moderate selectivity for CSC, and PepD displayed a bimodal response between subjects, with MMPi increasing activity from baseline. Salivary proteolysis of PepE was not effectively inhibited by either MMPi or COMi, as well as the combination thereof. The kinetic rates obtained for inhibited samples demonstrated correlated cleavage by different protease classes. The ratiometric analysis demonstrated relative MP/CSC cleavage in WMS of 1.8:1 (PepB), 1.36:1 (PepC), 0.96:1 (PepD), 0.88:1 (PepA) and 0.3:1 (PepE).

Together, residual levels of activity seen in inhibition profiles suggested substantial levels of active aspartate proteases, such as cathepsin D and lysyl-prolyl oligopeptidase, or microbial proteases such as thermolysin-like MPs or atypical U32 proteases, all of which are not inhibited by either inhibitor group.

### 3.2. Protease Profiling of Samples—Caries

The study next examined the enzyme activity derived from biopsy tissue homogenates to determine the protease activities and patterns of utilization within caries. Here, substrate cleavage was observed with effective utilization of substrates in terms of absolute turnover and relative to other substrates and inhibitor groups. Patterns of conserved, predominant proteolytic activity and inhibition types were observed (Figure 2A,B) before further categorization.

Within the biopsy homogenates, the relative and absolute activities observed were predominantly directed at PepC (40.4%, SD 16.1; average V_max_ = 723.0 min^−1^, SD 1013), PepE (24.2%, SD 13.4; average V_max_ = 481.6 min^−1^, SD 845.0) and PepA (15.8%, SD 8.2; average V_max_ = 163.1 min^−1^, SD 137.6), as shown in Figure 1. PepD (13.4%, SD 13.3; average V_max_ = 150.1 min^−1^, SD 261.8) and PepB (6.1%, SD 3.2; average V_max_ = 83.5 min^−1^, SD 97.9) were not consistently cleaved across caries samples, demonstrating different proteolytic enzymatic activity across lesions for these substrates.

Figure 2A highlights the differential enrichment within biopsies relative to saliva demonstrated caries-specificity of proteolytic fingerprints, despite substantial differences in sample dilutions between WMS and biopsy tissue, with enriched contributions of MP and CSC proteases in biopsy material. The caries samples demonstrated high MMP contributions directed at the substrates PepA (32.7% reduction, SD 21.9), PepB (20.9%, SD 12.2), PepE (15.8%, SD 16.2) and PepC (15.1%, SD 8.7). Together with direct correlation analyses (Figure 3A), this inferred a high joint MMP-2/MMP-8 contribution, as well as potential other MMPs activities such as MMP-14 and MMP-20, to the observed activity, with variable MMP-9-type activity. As with its contribution to overall protease activity in saliva, MMPs targeting pepD were variable in caries samples (11.0%, SD 10.5), thus inferring that MMP-2/predicted MMP-20 activities differed between samples.

For CSC proteases, cleavage rates positively correlated with those observed for MMPs for PepA and PepB, suggesting concomitant activity against given substrates by = CSC and MMP proteases in carious lesions, with the least contribution to activity by non-MMP/CSC proteases. Across caries biopsies, PepA was the most effective substrate processed at V_max_ for CSCs, with 34.5% inhibition of the kinetic activity attained, followed by PepD (23.9%, SD 23.3), PepB (22.1%, SD 19.9), PepE (14.8%, SD 21.6). Despite the highest kinetic turnover, PepC was not targeted by CSCs substantially (11.9%, SD 15.1). PepA displayed the greatest utilization consistently across all samples examined; in addition to its broad, pan-MMP cleavage, significant utilization by CSC classes evidenced effective and moderately specific utilization by these classes only. For peptides PepA, PepB and PepC, concomitant MMP and CSC activities were detected, BOTHi demonstrated an incomplete (but additive effect) relative to MMPi or COMi alone. This finding suggested that, in caries, both CSC and MMP types of proteases were active concomitantly within each lesion and did not affect activities of the protease classes present.

However, PepD and PepE samples, along with a subset of PepC samples, revealed the absence of an additive effect of BOTHi (i.e., greater inhibition than either MMPi or COMi alone), with inhibition efficiency for BOTHi lower than either of the highest MMPi or COMi inhibition rates. This observation suggested interactions between MMPs and CSC in caries, and may be due to inactive, respective other protease class interacting competitively with the substrate, as indicated by the altered turnover rates. Furthermore, it was observed that in many caries samples, the activity was increased rather than inhibited when inhibitors MMPi or COMi were present, relative to the non-inhibited control. These findings further supported the suggestions of complex interactions between the respective protease classes with direct effects on each other (activating or inactivating), or an indirect interaction, such as substrate binding (competitive or substrate/product inhibition) resulting in no cleavage taking place. Another potential explanation may be the effect of proteases on intrinsic inhibitors for protease classes.

Clustering proteolytic events in carious lesions identified two main proteolytic profiles clusters, predominantly based on relative PepE cleavage, irrespective of overall (total) protease activity levels. Clusters of activities were differentially delineated, as shown in Figure 2B and partial correlation in Figure 3A. These clusters are delineated by their inhibitable levels of MMP and CSC targeting PepE and PepA, suggesting MMP-2/MMP-8 and potentially MMP-20 (from cleavage predictions) activities. The second cluster was delineated by elevated CSC activities against PepB, PepC and PepD, with variable PepE CSC contributions.

### 3.3. Discrimination of Caries Status from Salivary Profiles

The disease specificity was assessed from the correlation between protease profiles in matched saliva and caries biopsy samples, and to determine if differentially affected fingerprint activities may be conserved in both specimen types. The differential utilization of peptides between caries and saliva samples highlighted those activities observed in the biopsy material were not predominantly derived from contaminating salivary activity. As evidenced by normalization for dilution factors, activities highly enriched within biopsy material commensurate with increased salivary concentrations. This demonstrated a highly active proteolytic activity in caries lesions examined and enabled inference of caries-specific profiles present in biopsy materials and matched saliva. The observation of such a direct relationship of fingerprint activities attributed salivary enrichment to biopsy activities. This finding demonstrated caries-specific, soluble enzymatic markers in saliva and directly linked fingerprint activity to the presence of a carious lesion.

Correlating matched specimen types and preference in substrate use, several proteolytic clusters were observed in caries and saliva (Figure 2A,B). The caries biopsy material revealed a significant proportion of MMP contribution in lesions, which corresponded with increased salivary MMP levels, although diluted with other protease profiles targeting given peptides. Furthermore, the network analysis conducted demonstrated that most differential features between caries status in saliva could be aligned to caries-specific interactions, as highlighted in the partial correlation network (Figure 3A–D).

As major features between caries states, saliva in caries-positive subjects is delineated predominantly by increased PepE overall turnover, BOTHi inhibition profiles and the reduced inhibitory effect of COMi and MMPi for this peptide. The second cluster in caries is defined by the prevalence of increased MMP contribution for PepA, PepB and PepE. The combined proteolytic activity was significantly elevated in the saliva of caries-positive individuals (*p* = 0.013 *, ES 0.78). For individual peptides, highly significant differences were observed for PepE activity (*p* = 0.005 **, ES 0.82; positive subjects: V_max_ 1500.2; negative: V_max_ = 601.8) and relative contribution of this peptide to overall proteolysis (*p* = 0.003 **, ES 0.95). This demonstrated an increase of PepE in absolute terms and disproportionately in relation to other substrates, mirroring biopsy material and defining caries-specificity of this substrate. In keeping with elevated proteolytic activities, PepC (*p* = 0.029 *, ES 0.7) and PepB (*p* = 0.016 *, ES 0.74) activities were elevated in absolute terms.

For protease families, PepB activities were significant for MMP (*p* = 0.017 *, ES 0.69), CSC (*p* = 0.018 *, ES 0.72) and dual inhibition (*p* = 0.025 *, ES 0.64). PepA was significant for BOTHi inhibition (*p* = 0.025 *, ES 0.64). PepE in caries-positive individuals demonstrated but increased MMP activity (*p* = 0.05 *).

Aside from the altered proteolytic profiles delineating caries presence, the prevalence of disinhibition effects (where inhibited groups had greater activity than the uninhibited control) were commonly observed. This effect was encountered for PepE COMi/BOTHi for caries-negative saliva. This consistent effect suggested an inhibitory effect of the CSC proteases themselves on proteases not inhibited by BOTHi. This effect was more pronounced in saliva derived from caries-positive individuals for the other substrates. Here, PepD and PepE MMPi increased overall activity beyond the positive, uninhibited control material. For PepE, this effect was not observed in the biopsy material, suggesting interactions with salivary, non-caries proteases with caries derived MMP proteases. In PepC, all inhibitor groups (MMPi, COMi and BOTHi) demonstrated substantial subclusters were disinhibited in saliva. For these cases, the effect was correlated to the biopsy material. This finding implies an effect of BB-94 inhibitable proteases on non-MMP activities in disease phenotypes. When substantive MMPi inhibition was observed for PepA/PepB in any sample (>25%), the PepA or PepB COMi turnover was consistently higher than the control activity, and the BOTHi group did not attain inhibition levels seen in MMPi (i.e., greater activity in the combined group, than in the individual inhibition groups). This suggested a network of conditional agonistic interactions between proteases in caries, or a sequential, cascading activation.

Here, the contribution of MMP targeting PepA and the combined inhibition (BOTHi) and a positive CSC contribution to PepB and PepE cleavage and PepE absolute activity were indicators for caries presence. Increases in overall PepA and PepC activity were indicators of no caries being present, as was MMP activity targeting PepE. In caries-negative individuals, little MMPi inhibition was observed across all patients. In contrast, the PepE COMi treatment displayed increased enzymatic activity than the control (no treatment with inhibitor), maintained but not further amplified in the PepE BOTHi group. This observation was associated with the presence of significant MMP levels, as indicated by the broad-range, high activity peptides PepA and/or PepB, suggesting that MMPs present in the sample exerted an indirect effect on proteolytic cleavage of PepE via a non-CSC or MMP class protein. Furthermore, where relatively low, overall kinetic activities were observed in the tissue material, the increasing kinetic activity here was driven by MMPs targeting PepA, PepB and PepC. In caries positive individuals, this was reversed to MMPi, increasing the kinetic activity relative to the control, with significant CSC (COMi) inhibition activity frequently observed; however, this effect was ablated in the BOTHi group. In healthy individuals, the interplay between CSC and MMPs was highlighted in peptides A, B and C (and to a lesser degree D) by the BOTHi group. Whilst inhibitions were noted for either MMPi, COMi or both, the combination with BOTHi reduced the efficacy to levels below the individual inhibitor class, i.e., the residual kinetic activity when BOTHi was administered was greater than the respective MMPi and/or COMi alone. These findings demonstrated an interaction of MMP/CSC either with the substrate complex without cleavage or the ablation of proteolytic events, directly affecting the effector protease. These effects were evident within the caries biopsies.

### 3.4. Effects of Confounding Factors

To ascertain the specificity of the effects and potential confounders observed, pairwise and confirmatory regression analyses between factors was conducted. Factors between groups were the age of patients within the caries group, the prevalence of periodontal disease, social factors, and the presence of relevant co-morbidities in relation to absolute and relative protease levels. Age was correlated with increases in PepE absolute, but not relative, levels (*p* = 0.02 *, r = 0.347), as well as salivary protein concentration (*p* = 0.003 **, r = 0.6). Further positive correlations with age were the identified confounders medication and cardiovascular disease (*p* = 0.001 ***). Female gender was associated with lower caries protein content (*p* = 0.05 *, ES 0.7), PepC MMPi relative activity loss (*p* = 0.02 *, ES 0.7). Based on gender, caries presence and severity were lower in the study group (*p* = 0.03 *). In asthmatic patients taking inhaled or oral steroids, PepC absolute activity (*p* = 0.02 *, ES 0.7), PepD relative contribution (*p* = 0.02 *, ES 0.7) and its MMP/CSC ratio (*p* = 0.004 ***, ES 1.6) was significantly altered. A positive substance abuse status (*n* = 9), correlated with an increased degree of MMPi inhibition attained for PepA (*p* = 0.02 *, ES 0.7), PepC MMP/CSC ratio (*p* = 0.015 *, ES 0.95) and PepE CSC inhibition (*p* = 0.01 **, ES 0.9), PepB BOTHi inhibitions for dual inhibition were altered in salivary and biopsy specimen (S: *p* = 0.038 *, ES 0.8; C: *p* = 0.013 *, ES 1.7). Active periodontal disease cases demonstrated significantly elevated absolute turnover of PepA BOTHi (*p* = 0.028 *), PepC (*p* = 0.035 *, ES 0.75) and PepC MMPi (*p* = 0.011 *, ES 0.92), PepB total activity (*p* = 0.025 *) as well as for PepB COMi (*p* = 0.044 *, ES 0.69), however, this did not affect relative levels or any feature of the biopsy material.

### 3.5. Classification Using Supervised Machine Learning and Correlation

To determine the applicability of profiles observed to detect caries, after the initial analysis of descriptive statistics, a sequential analysis supported by supervised machine learning was carried out. Logistic regression was conducted utilizing cleavage rates of all five substrates and their inhibitor profiles. After feature selection and accounting for potential over-fitting of the model, a good initial fit of parameters selected was demonstrated (AIC 50.1, BIC 67.1, Nagelkerke R2 0.691). Classification prediction provided a sensitivity of 0.90 and a specificity of 0.85, suggesting a high sensitivity with the selected parameters, and the logistic regression model was statistically significant (χ^2^ (29) 22.5, *p* ≤ 0.001 ***). The model correctly classified 93.1% of all cases. A sensitivity and specificity of 84.6% and 90.0%, respectively was attained (AUC 0.94, precision 0.93, F-measure 0.92). Misclassifications were observed in 2/13 negative (15.4%) and 3/27 (10%) caries-positive saliva samples, giving overall false-positive and false-negative readouts of approximately 5% in 7% of the cases examined, respectively. Using markers highlighted, random forest classification, and a tree algorithm with forward pruning demonstrated a clear delineation of positive and negative caries status (Figure 3B). These results demonstrated the feasibility of this approach in categorizing caries diseases state from substrate cleavage signatures. Using PepE and inhibitor groups as a single peptide to detect caries, improved positive classification could be carried out, identifying 81.8% of caries patients, with a precision of 80.6% and a sensitivity of 90.6% (AUC 0.815), but at significantly lower, prevalence-unweighted specificity of 50%.

## 4. Discussion

The present study examined the feasibility of using arrayed protease activity assays to delineate generic destructive processes by proteases, rather than specific and divergent enzyme identities of effector proteins as ‘*metabiomarkers*’ of oral disease, with specific utility in caries diagnosis. To date, no single biomarker has been described to determine the caries status of individuals, owing to the highly diverse and changing microbial communities within caries lesions and the oral environment, as well as the intrinsic lability of salivary proteins after sampling due to proteolytic degradation from a variety of different proteases present [15,24,25]. Within the complex salivary protease networks observed, individual time-lapse measurement derived from individual, moderately specific FRET-substrates can relate to disease processes without the need to resolve specific proteases. This study utilised the inference of biological activity from panels of FRET-substrates, probes designed based on known natural targets of gelatinases and collagenases, or those able to cleave the unfavourable, hydrophobic, or bulky sites indicative of resistant ECM molecules. Whilst significant inhibition was attained with metalloprotease inhibitors for some substrates with specificity to caries, suggesting enrichment for MMP-2, MMP-8, MMP-9 and potentially MMP-20 were observed, no complete inhibition was achieved with either combination of protease inhibitors, in keeping with previous findings [14].

Furthermore, the contribution of MMP-8, from periodontal disease activity [26] likely reduced the resolving power of caries profiles. However, the findings did highlight the complex and networked proteolytic interactions in saliva and caries biopsies and the dysregulation of these in disease states. The combined use of multiple substrates and inhibitors in the present study suggests that the combination of inhibition groups enabled the prediction of multiple active protease classes in carious lesions, demonstrating a combined contribution of different proteolytic entities to tissue destruction. The finding of combined MMP and likely cathepsin activity suggests sequential processing of structural dentine proteins, as observed in the interplay between MMPs and cathepsin in osteoclastic bone remodelling [27]. At an estimated population prevalence of 20%, such approaches may yield false positives in 8% of cases, provide true-negative rates of 0.75 and a population level false-positive rate of 0.1, highlighting the potential utility of such MLS-derived approaches in the detection of oral disease, particularly if these are used to adjunct, rather than replace, periodic full dental examinations. Thus functional, pathological protease activities arising from the saliva sample were determined and could be differentiated from profiles obtained from caries-free individuals with high levels of confidence and differentiated from other oral conditions such as periodontal disease.

Enzyme kinetic activity assays demonstrated that non-MMP activities predominate, in addition to the previously described human metalloproteases described in lesions, and WMS profiles. The proteolytic differential profiles obtained supported the hypothesis that one or more pathognomonic protease activities measured in saliva are present in caries-positive subjects, and the enrichment within the biopsy material placed these activities as arising from the caries process. Diverging protease activities within the lesion were observed, as indicated by changes in the substrate utilization and protease classes covered by other substrates used in the panel, thereby increasing the sensitivity of the assay, as well as the caries-specificity.

A surprising finding was the ‘disinhibition’ of proteolytic activity by the addition of class-specific protease inhibitors, even when high catalytic rates were observed. This phenomenon was encountered predominantly in caries saliva and tissue material. These effects were not mitigated by the increased spectrum of proteases covered the dual inhibition group, potentially suggesting multi-family protease-protease interactions. These gain-of-function activities merit further investigation in terms of protease-protease interactions and their potential roles in disease progression.

The study, despite the clear distinction of disease states using this method, has limitations due to the patient cohort studied, the study design and some unexpected findings from suspected protease-protease interactions. The limited and heterogenous patient cohort recruited, whilst deliberately chosen to reflect the presence of multiple oral and systemic conditions often encountered, had significant confounders, also preventing greater differential resolution by protease fingerprints. This is exemplified by PepA (a high turnover substrate of MMP-8) failing to statistically resolve disease cases, despite elevated levels. This is likely due to MMP-8 being an indicator of both, caries and active periodontal disease [26,28,29] Furthermore, the heterogeneity of the patient cohort, were predominantly advanced disease stages and further studies are warranted to ascertain detection thresholds.

In addition to supporting this biological activity-based meta-biomarker approach, the findings did, however, highlight the diversity of protease families targeting collagenolytic/gelatinolytic substrates, many of which are attributed with MMP specificity, and supported the presence of protease-protease interactions affecting cleavage of substrates, and likely their biological counterparts. Furthermore, to assess the clinical applicability of proteolytic fingerprinting, patients with combined and/or advanced disease states were deliberately included, but this limits inference of detection thresholds in earlier disease and may have reduced the potential resolution from proteases differentially expressed in inflammatory states, together with potential confounding arising from medical conditions or substance abuse issues but represent patients with increased oral health needs and more likely to experience adverse sequelae of oral disease. For these patients, underlying conditions or pharmacological treatments frequently affect oral function, such as saliva flow [30,31], thus altering protease quantities and activities.

The lack of exclusion of such confounders was overcome by the causal inference of activity arising from the biopsy material, and carry-over observed in saliva samples, irrespective of oral or systemic disease. The altered profiles for the periodontal status align with previous findings of host proteases, namely MMP-2. MMP-8 and MMP-9, as well and bacterial gingipains (targeted by COMi), were identified in the periodontal, non-caries group and the combined periodontal and caries cohort which may have reduced the sensitivity of the assay to detect the caries-specific activity of MMP-8 and cathepsins previously described. Whilst fluorescence-based measurements may not be easily implemented within a clinical setting, colorimetric protease measurements have been validated [32] and thus would facilitate use of protease profiling, and further supported by the analysis of select peptidases herein to demonstrate sensitivity to, and discrimination of, oral diseases.

With a clinical use in mind, this study explored the potential utility of implementing a condensed probe set for this protease activity-based approach. Given the individual response, without other peptides, PepE displayed good stand-alone caries-specificity, based on activity. The inclusion of inhibitors in parallel to this single peptide enables improved sensitivity to caries and to identify periodontal phenotypes, likely due to ADAM8 metalloprotease activity as previously described [33]. Although our results provide pilot data for the functional and diagnostic role in salivary profiling of protease networks in oral diseases, more extensive studies will be required to validate this approach. Nevertheless, the present methodology may represent a feasible, top-down approach to elucidate the complex interplay of proteases within complex, poorly characterized disease states. Further studies will assist in refining and validating the approach, define detection limits for earlier and other oral diseases and delineate interactions arising from concomitant systemic disease and treatments.

## 5. Conclusions

The present data provide proof-of-concept for the applicability of multiplexed protease substrates for the facile, rapid, and sensitive detection of functional proteolytic markers of ongoing tissue destruction in saliva. The findings highlighted the complex contribution of host and microbiome-derived proteases derived from different classes, as well as complex protease-protease interactions. The use of a target-driven approach for screening or as an adjunctive tool may assist in improved and targeted prophylactic, diagnostic and treatment provisions and holds the potential to provide measures of disease susceptibility or activity present.

## Figures and Tables

**Figure 1 jpm-11-00866-f001:**
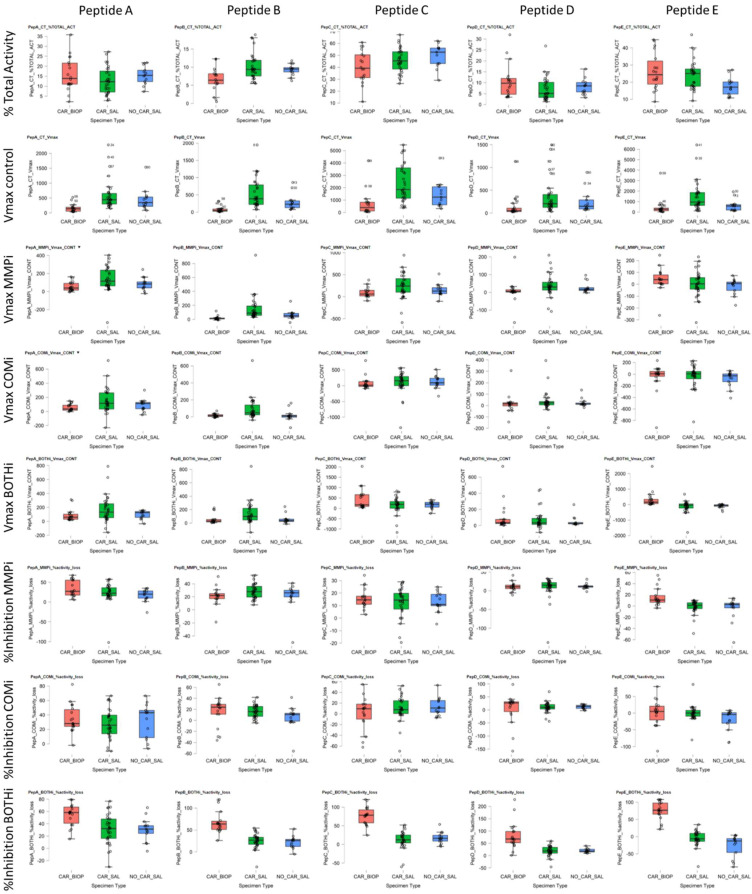
Boxplots for absolute (Vmax) and substrate turnover of peptides used and inhibitors (Vmax), relative (% TOTAL-ACT) substrate utilization across peptides (Peptides **A**–**E**), and contribution of inhibitor-doped samples (% Inhibition). Sample types were WMS, based on caries status (CAR/NO-CAR), and caries biopsy (BIOP) samples for the five peptides studied. Inhibitor data is expressed as Vmax contribution to signal and % inhibition by either Metalloproteases (MMPi), CSC proteases (COMi) or combined MMPi and COMi (BOTHi). relative (%CONT; percentage contribution to turnover; %Inhibition: Percentage inhibition of overall activity).

**Figure 2 jpm-11-00866-f002:**
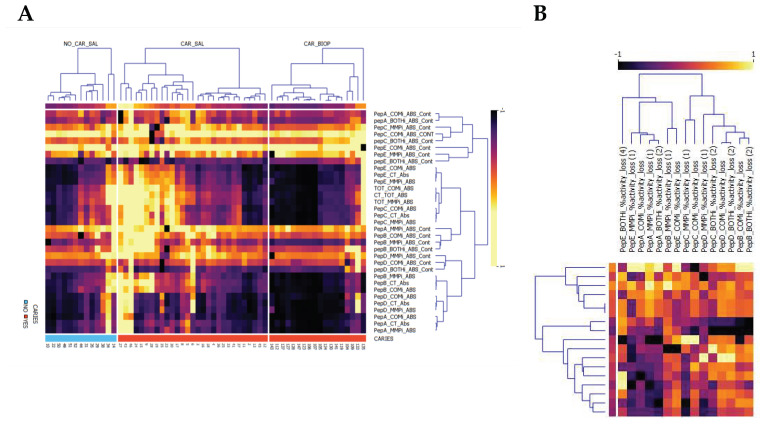
Normalized Pearson Bi-clustered heatmaps, (**A**) split by sample type for all absolute activities and (**B**) inhibitor contributions to turnover. Specimen were saliva (_SAL) samples examined and caries biopsies obtained (CAR_BIOP), representing relative substrate utilization for peptides (PepA–PepE) turnover in relation to overall activity (_CT_ABS) and inhibited contribution to turnover (_ABS_CONT). Inhibitors denote _COMi_ = COMPLETE, _MMPi = Metalloprotease inhibitor or _BOTHi, comprising a combination of both. The respective contribution of inhibitors (CONT) to Vmax is reported. Across all samples, a consistent CSC signal is observed for PepC (PepC_COMi_ABS_CONT). Saliva in caries positive subjects is delineated predominantly by an increase in pepE overall turnover, BOTHi inhibition and the reduced inhibitory effect of COMi and MMPi. The second cluster in caries is defined by the prevalence of increased MMP contribution for PepA and PepB. (**B**) relative inhibition observed for caries biopsies, highlighting the utilization of substrates by both MMP and CSC proteases.

**Figure 3 jpm-11-00866-f003:**
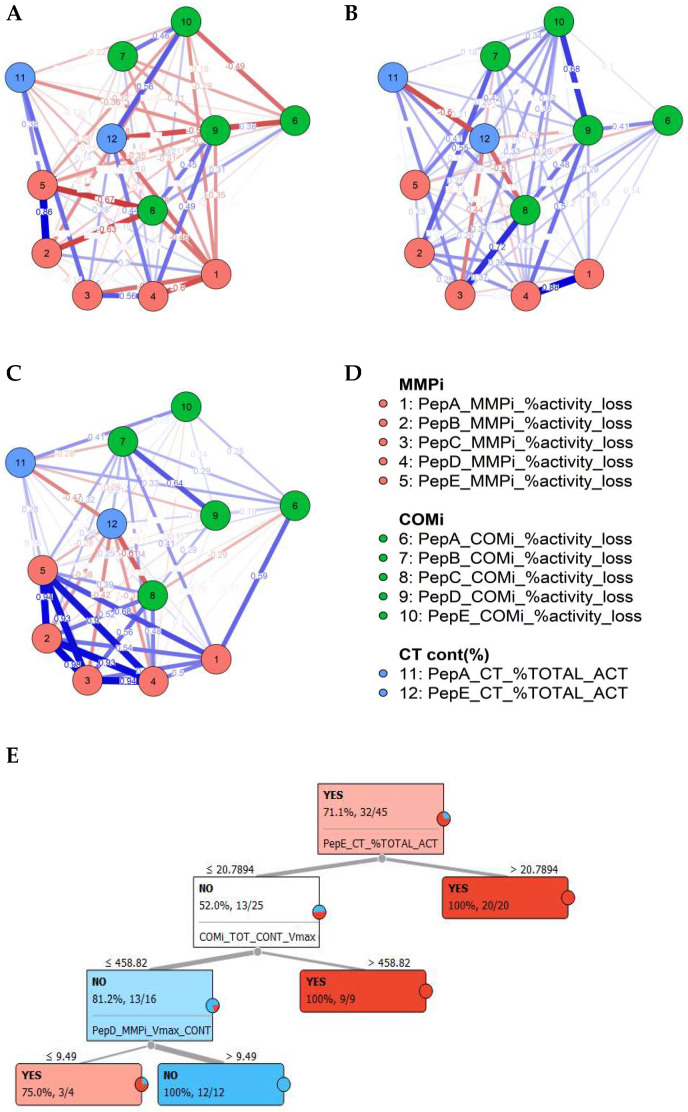
Delineation of caries status, based on the proteolytic fingerprints. A–D, Partial correlation demonstrating weighted correlations of inhibitors and activities between saliva samples from patients based on caries status as no caries saliva (**A**), caries saliva (**B**) and caries biopsy samples (**C**) and indicating conserved differences in salivary profiles, correlating with the biopsy material (**A**–**D**); Random Forest decision tree for salivary caries determination (YES/NO) for distribution-centred MLA classifier analysis (**E**).

**Table 1 jpm-11-00866-t001:** Clinical data on dental patient cohort used in this study (*n* = 52).

Patient Characteristics	
Patient number	52 (45 used for downstream analysis)
Mean Age (years, range)	52 years (range 19–84)
Male/Female (%)	52/48
Dentine caries present/absent	32/13
Caries severity (*n*; mild and moderate, severe) *	24/8

* Caries severity was assessed as described in Section 2.1.

**Table 2 jpm-11-00866-t002:** Predicted and known cleavages rates, sites and effector protease identities for multiplexed peptides pepA–pepE used in this study. Predictions were derived from [21]. Cleavage rates adapted taken from [22,23].

Peptide ID	Sequence	Sequence Used for Query	Aspartic	Cysteine	Metallo	Serine	Known Cleavage & Rates (M-1 S-2)	Predicted Cleavage Identity
PepA (PEPDAB005)	Dabcyl-Leu-Ala-Gln-Ala-(Homo phenylalanine)-Arg-Ser-Lys(5-FAM)-NH_2_	LAQAFRSK		X QAFR|SK	X LAQA|FRSK	X AQAF|RSK	MMP2 (3.2 × 10^5^) MMP8 (1.4 × 10^5^) MMP9 (2.2 × 10^5^)	Cys: cathepsin K (C01.036) Met: matrix metallopeptidase-9 (M010.004) Ser: chymotrypsin A (cattle-ty)pe)
PepB (PEPDAB008)	Dabcyl-Pro-Cha-Gly-Cys(Me)His-Ala-Lys(5-FAM)-NH_2_	PLGCHAL			X PLG|CHAL		MMP2 (2.9 × 10^4^) MMP8(2.4 × 10^4^) MMP9 (8.5 × 10^5^)	Met: matrix metallopeptidase-2 (M10.003)
PepC (PEPDAB010)	Dabcyl-Ser-Pro-Leu-Ala-Gln-Ala-Val-Arg-Ser-Ser-Lys(5-FAM)-NH_2_	SPLAQAVRSSK		X QAVR|SSK	X LAQA|VRS	X AQAV|RSSK &SPL|AQAV	MMP2 (1.7 × 10^5^) MMP8 (2.6 × 10^4^) MMP9 (6.0 × 10^5^)	Cys: cathepsin K (C01.036) Met: matrix metallopeptidase-9 (M10.004) Ser: elastase-2 (S01.131)/cathepsin G(S01.133) Other: proyl peptidase
PepD (PEPDAB013)	Dabcyl-His-Gly-Asp-Gln-Met-Ala-Gln-Lys-Ser-Lys(5-FAM)-NH_2_	HGDQMAQKSK		X HGDQ|MAQK	X HGDQ|MAQK		MMP2 (2.4 × 10^3^) MMP8 no activity MMP9 no activity	Cys: cathepsin K (C01.036) Predicted MMP20 activity
PepE (PEPDAB014)	Dabcyl-Glu-His-Ala-Asp-Leu-Leu-Ala-Val-Val-Ala-Lys(5-FAM)-NH_2_	EHADLLAVVAK	X DLLA|VVAK		X EHAD|LLAV	X ADLL|AVVA	MMP2 (6.3 × 10^3^) MMP8 (4.8 × 10^3^) MMP9 no activity	Asp: cathepsin D/cathepsin E Met: matrix-metallopeptidase-2 (M10.003), Ser: cathepsin G (S01.133)

## Data Availability

All data is included within the manuscript.

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
