# Peer review of "Clinical Evaluation of Pathognomonic Salivary Protease Fingerprinting for Oral Disease Diagnosis"

_jpm, 2021, doi:10.3390/jpm11090866_

Round 1

Reviewer 1 Report

All names of enzymes should be mentioned in the manuscript with the small first letter. Please correct. 

There has not been described the method for obtaining saliva samples. Please provide. 

The authors of the manuscript didn't mention weak spots of this pilot study, like missing the control group. It is part of further validation studies that have been mentioned, but should me stated clearly. 

Author Response

All names of enzymes should be mentioned in the manuscript with the small first letter. Please correct. 

Authors' response: we have corrected all enzyme names accordingly. 

There has not been described the method for obtaining saliva samples. Please provide. 

Authors' response: Thank you for this important advice. We have now included a separate section in M&M part 2.1. (Line 117-123 of the revised manuscript) to describe our method for saliva collection. 

The authors of the manuscript didn't mention weak spots of this pilot study, like missing the control group. It is part of further validation studies that have been mentioned, but should me stated clearly.

Authors' response: We have now included a  statement stating the limitations associated with our study (line 477-486) and highlighted the necessity to perform further validation studies. 

Reviewer 2 Report

v

Dear authors,

Thank you for the opportunity to review your manuscript. The aim of the presented research article was to examine the feasibility of the diagnostic use of multiplexed, FRET-based protease substrates to delineate proteolytic networks in oral disease. The article is interesting. The problem is actual and very well presented, the introduction and discussion are reliable and comprehensive with an actual review of the literature. The manuscript ends with the appropriate conclusions. However, I recommend some minor revisions:

Material and Methods section:

Table 1 -age -52 years-is it median or mean age?

Results

Lines 239 and 303 please paraphrase the sentences-they shouldn’t start with “we”

Discussion

I think that the author should add a short paragraph about limitations of the study

What is a practical application of this study?

Author Response

We thank the reviewer for positive evaluation and helpful comments on our manuscript. We address the comments accordingly:

Material and Methods section:

Table 1 -age -52 years-is it median or mean age?

Authors' response: This is the mean age. We have now mentioned this fact in the table 1. 

Results

Lines 239 and 303 please paraphrase the sentences-they shouldn’t start with “we”

Authors' response: We have now rephrased these sentences to avoid "we". 

Discussion

I think that the author should add a short paragraph about limitations of the study

Authors' response:  Thanks for this helpful comment. We have now included a chapter in the discussion (line 477-486) describing the limitations of our study. 

What is a practical application of this study?

Authors's response: we strongly believe that our study can be a profound basis for the development of a reliable first-line test to stratify oral diseases. Given our pilot data, we will explore our strategy on a larger patient cohort to validate our findings.